# Development of the RF-GSEA Method for Identifying Disulfidptosis-Related Genes and Application in Hepatocellular Carcinoma

**Linghao Ni** [†] [ID]**, Qian Yu** [†]**, Ruijia You, Chen Chen and Bin Peng** *

School of Public Health, Chongqing Medical University, Chongqing 400016, China
* Correspondence: pengbin@cqmu.edu.cn; Tel.: +86-139-8305-9001
† These authors contributed equally to this work.

**Abstract:** Disulfidptosis is a newly discovered cellular programmed cell death mode. Presently, a considerable number of genes related to disulfidptosis remain undiscovered, and its significance in hepatocellular carcinoma remains unrevealed. We have developed a powerful analytical method called RF-GSEA for identifying potential genes associated with disulfidptosis. This method draws inspiration from gene regulation networks and graph theory, and it is implemented through a combination of random forest regression model and Gene Set Enrichment Analysis. Subsequently, to validate the practical application value of this method, we applied it to hepatocellular carcinoma. Based on the RF-GSEA method, we developed a disulfidptosis-related signature. Lastly, we looked into how the disulfidptosis-related signature is connected to HCC prognosis, the tumor microenvironment, the effectiveness of immunotherapy, and the sensitivity of chemotherapy drugs. The RF-GSEA method identified a total of 220 disulfidptosis-related genes, from which 7 were selected to construct the disulfidptosis-related signature. The high-disulfidptosis-related score group had a worse prognosis compared to the low-disulfidptosis-related score group and showed lower infiltration levels of immune-promoting cells. The high-disulfidptosis-related score group had a higher likelihood of benefiting from immunotherapy compared to the low-disulfidptosis-related score group. The RF-GSEA method is a powerful tool for identifying disulfidptosis-related genes. The disulfidptosis-related signature effectively predicts HCC prognosis, immunotherapy response, and drug sensitivity.

**Keywords:** disulfidptosis; random forest regression model; hepatocellular carcinoma

## 1. Introduction

Disulfidptosis is a newly identified mode of regulated cell death mediated by the cystine transporter solute carrier family 7 member 11 (SLC7A11). This mode is characterized by the collapse of cytoskeletal proteins and F-actin due to the accumulation of intracellular disulfide bonds [1]. The previous study further indicates that glucose transporter inhibitors induce disulfidptosis, subsequently regulating tumor proliferation, highlighting the significance of disulfidptosis in cancer therapy. SLC7A11, the central gene of disulfidptosis, enhances glutathione (GSH) production. This mitigates cell death induced by oxidative stress, including ferroptosis, by importing cystine [2]. SLC7A11 is a double-edged sword in redox regulation. Although SLC7A11 is widely acknowledged for its established role in promoting cell survival by inhibiting ferroptosis, emerging research has uncovered unexpected implications. Under conditions of glucose deficiency, SLC7A11 exerts a counterintuitive role in facilitating cell death by promoting disulfidptosis [3,4]. Preclinical study findings indicate that the utilization of glucose transporter (GLUT) inhibitors as part of metabolic therapy can induce disulfidptosis and impede cancer growth [5]. The research on disulfidptosis is currently in its early stages, with many genes involved in this process yet to be identified and the specific biological mechanisms remaining unclear. There is an

urgent need for the identification of genes associated with disulfidptosis to facilitate further investigation into this phenomenon.

Liver cancer ranks as the sixth most prevalent malignancy on a global scale and stands as the fourth primary contributor to cancer-associated mortality, demonstrating a notably low 5-year survival rate of 18% [6,7]. Hepatocellular carcinoma (HCC) represents the most predominant histological subtype (90%) of primary liver cancer and concurrently exhibits the highest mortality rate among all categorizations [8]. Because of the highly insidious nature of HCC, most patients are already in the middle to late stages of being diagnosed. It is also highly heterogeneous and progresses rapidly, making clinical treatment extremely difficult. Current treatments for early-stage HCC include surgery (liver resection or liver transplantation), radiofrequency ablation, radiotherapy, and intra-arterial therapies [9]. For patients with advanced-stage HCC, various systemic therapies are their best option. Targeted anti-tumor agents (TKIs), sorafenib and lenvatinib, are the first-line drugs used clinically for the treatment of advanced-stage HCC [10,11].In recent years, immune checkpoint inhibitors (ICIs) have brought about a transformative paradigm shift in cancer treatment and have garnered heightened attention for their application in addressing HCC. Many studies have shown that the efficacy of ICIs, alone or in combination with ICIs and TKIs, in the treatment of HCC has surpassed that of traditional first-line therapeutic drugs [12,13]. Although researchers have made great strides in the field of immunotherapy, the response rate to immunotherapy in HCC patients is only 15–20% [14]. There is an urgent need to develop a signature to predict prognosis and immunotherapy response in HCC. Although some studies have demonstrated the value of disulfidptosis in the development and prognosis of lung adenocarcinoma, breast cancer, and renal cell carcinoma [15–17], the specific mechanism of disulfidptosis and its actual value in the development and prognosis of HCC have not yet been clearly revealed.

In this study, we have developed a method based on the random forest regression model and Gene Set Enrichment Analysis (GSEA) to identify potential disulfidptosis-related genes. We applied this method to hepatocellular carcinoma to demonstrate its practical utility and application value. Based on the genes identified using this method, we performed Consensus Clustering analysis, Lasso-Cox regression, and Gene Set Variation Analysis (GSVA) to construct a disulfidptosis-related signature. Subsequently, we conducted an in-depth assessment of the value of this signature in individualized therapy in HCC patients.

## 2. Materials and Methods

### 2.1. Data Sources and Processing

The pan-cancer RNA-seq data and the corresponding clinical information were downloaded from the UCSC Xena data portal (https://xenabrowser.net/ (accessed on 23 September 2023)). HCC patients' RNA-seq data and corresponding clinical information were sourced from the TCGA (https://xenabrowser.net/ (accessed on 23 September 2023)), GEO (https://www.ncbi.nlm.nih.gov/gds/ (accessed on 23 September 2023)), and ICGC (https://dcc.icgc.org/ (accessed on 24 September 2023)) databases. After excluding suspected non-HCC samples, duplicate samples, samples lacking clinical information, samples with a survival time of less than one month, and samples subjected to immunotherapy, the TCGA-LIHC dataset comprised 322 samples, the GSE116174 dataset comprised 64 samples, and the LIRI-JP dataset comprised 169 samples (Table 1). In addition, we obtained transcriptomic data and corresponding clinical information from a cohort (IMvigor210) receiving anti-PD-L1 immune therapy using the "IMvigor210CoreBiologies" package [18]. A total of 84 disulfidptosis-related genes were obtained from the FerrDb database (http://www.zhounan.org/ferrdb/current/ (accessed on 28 September 2023)) [19]. Three RNA-seq cohorts (TCGA-LIHC, LIRI-JP, and IMvigor210) were normalized using log2 (TPM + 1), while one Affymetrix microarray cohort (GSE116174) was normalized using log2 (RMA + 1).

**Table 1.** Clinical information of three datasets.

| Variable | | TCGA-LIHC (*n* = 322) | GSE116174 (*n* = 64) | LIRI-JP (*n* = 169) |
|---|---|---|---|---|
| **Age** | | | | |
| | ≤60 | 154 (47.8%) | 46 (71.9%) | 36 (21.3%) |
| | >60 | 168 (52.2%) | 18 (28.1%) | 133 (78.7%) |
| **Gender** | | | | |
| | Female | 100 (31.1%) | 6 (9.4%) | 40 (23.7%) |
| | Male | 222 (68.9%) | 58 (90.6%) | 129 (76.3%) |
| **TNM_Stage** | | | | |
| | I | 170 (52.8%) | 7 (10.9%) | 30 (17.8%) |
| | II | 69 (21.4%) | 1 (1.6%) | 82 (48.5%) |
| | III | 80 (24.8%) | 45 (70.3%) | 43 (25.4%) |
| | IV | 3 (0.9%) | 11 (17.2%) | 14 (8.3%) |
| **Grade** | | | | |
| | G1 | 50 (15.5%) | - | - |
| | G2 | 149 (46.3%) | - | - |
| | G3 | 111 (34.5%) | - | - |
| | G4 | 12 (3.7%) | - | - |
| **Status** | | | | |
| | Alive | 208 (64.6%) | 37 (57.8%) | 143 (84.6%) |
| | Dead | 114 (35.4%) | 27 (42.2%) | 26 (15.4%) |
| **Alcohol** | | | | |
| | No | - | 51 (79.7%) | - |
| | Yes | - | 13 (20.3%) | - |
| **Smoke** | | | | |
| | No | - | 33 (51.6%) | - |
| | Yes | - | 31 (48.4%) | - |
| **Invasion** | | | | |
| | No | - | 35 (54.7%) | - |
| | Yes | - | 29 (45.3%) | - |
| **HBV** | | | | |
| | No | - | 17 (26.6%) | - |
| | Yes | - | 47 (73.4%) | - |

*2.2. The RF-GSEA Method*

The fundamental concept of the RF-GSEA method is to amalgamate machine learning and enrichment analysis to aid in comprehending the associations among genes within the gene regulatory network (GRN) and assessing whether they are linked to specific biological processes or pathways. The mutual regulatory relationships among genes form a complex GRN that plays a crucial role in biology and medicine. Genes involved in the same biological process exhibit tighter connections within the GRN [20]. In graph theory, such a phenomenon can be described as a subgraph within a complex network [21]. From this, we can infer that if a gene participates in a specific biological process, it should exhibit regulatory relationships within the gene regulatory network with other genes that are also involved in the same biological process. Furthermore, the greater the number of regulatory relationships a gene has with other genes involved in the same biological process, the higher the probability that it is indeed a participant in that biological process. We assume that the actual regulatory relationships of some disulfidptosis-related genes are depicted as shown in Figure 1A. Our method requires some known disulfidptosis-related genes as prior information genes. The FerrDb database is a specialized database focused on programmed cell death, gathering a considerable number of related genes [19]. Based on current research on disulfidptosis, the FerrDb database has compiled a total of 84 genes related to disulfidptosis (Table S1). The FerrDb team has classified them into three groups based on their current level of confidence: validated, screened, and deduced (Figure 1B). These 84 genes are not all definitively disulfidptosis-related genes, and as a result, the network they form may differ from the actual network. These 84 genes were used as prior

information genes for the RF-GSEA method. The random forest model is an excellent machine learning algorithm, and numerous researchers have utilized its importance scores for the construction of GRN [22,23]. In gene regulatory networks constructed using the random forest method, if a significant proportion of nodes with a high likelihood of being connected to gene *n* consist of pre-defined prior information genes, then gene *n* is highly likely to be associated with disulfidptosis (Figure 1C). GSEA is a method that allows for enrichment analysis based on gene ranking and scores [24]. We employ it to assess whether a significant proportion of genes with a high likelihood of being connected to gene *n* are pre-defined prior information genes.

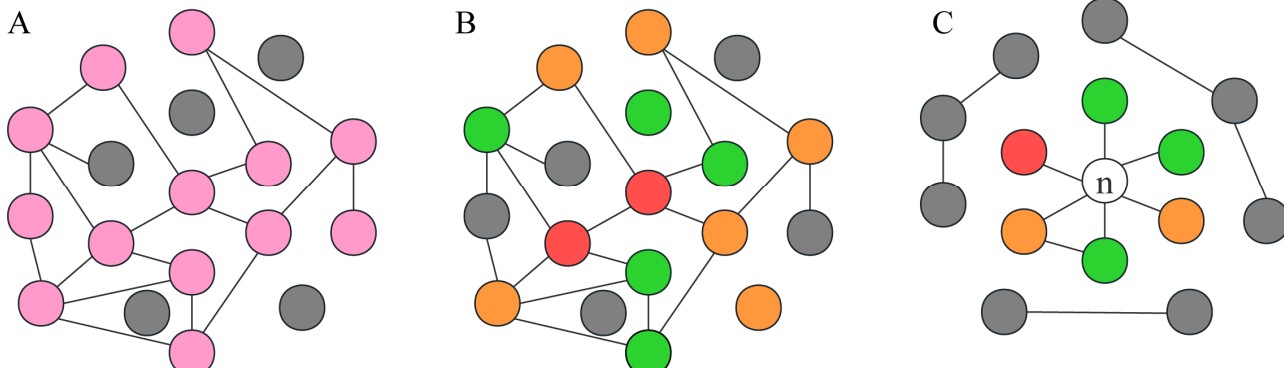

**Figure 1.** The RF-GSEA method relies on gene regulatory networks and graph theory. (**A**) GRN of disulfidptosis-related genes. Pink circles represent disulfidptosis-related genes, while gray circles represent non-disulfidptosis-related genes. (**B**) GRN of 84 disulfidptosis-related genes from the FerrDb database. The red, orange, and green circles, respectively represent three different confidence levels: validated, screened, and deduced. (**C**) GRN of a disulfidptosis-related gene *n* and disulfidptosis-related genes from the FerrDb database.

To summarize, the RF-GSEA method determines if a gene *n* is associated with disulfidptosis by following four specific procedures. Firstly, we obtained RNA-seq data of HCC patients from the TCGA database and performed log2 (TPM + 1) transformation. Subsequently, we used gene *n* ($n \in N$) as the dependent variable and all other N-1 genes as independent variables in a random forest regression model (Figure 2A). Secondly, for each random forest regression model, we constructed 500 Classification and Regression Trees (CART). Each tree was built by examining $\sqrt{N-1}$ features at each node, without applying any pruning techniques, thus allowing them to expand without restrictions (Figure 2B). Thirdly, the random forest scores of the independent genes are defined as the increase in mean squared error (%IncMSE) or increase in node purity (IncNodePurity) after their random replacement. After that, we ranked all genes (N-1) in descending order based on their random forest scores and conducted GSEA analysis (Figure 2C). Finally, we downloaded 84 disulfide death-related genes from the FerrDb database to construct the gene set for GSEA. If the ranked genes could achieve significant enrichment ($p < 0.05$), then gene *n* could be considered a candidate disulfidptosis-related gene (Figure 2D). Repeating the previous four steps, iterate through all N genes to identify disulfidptosis-related genes. To enhance the robustness of our results, we intersected the enrichment results from the two random forest score methods (%IncMSE and IncNodePurity) with the DEGs identified by "limma" analysis. This intersection served as the set of disulfidptosis-related genes for subsequent analyses.

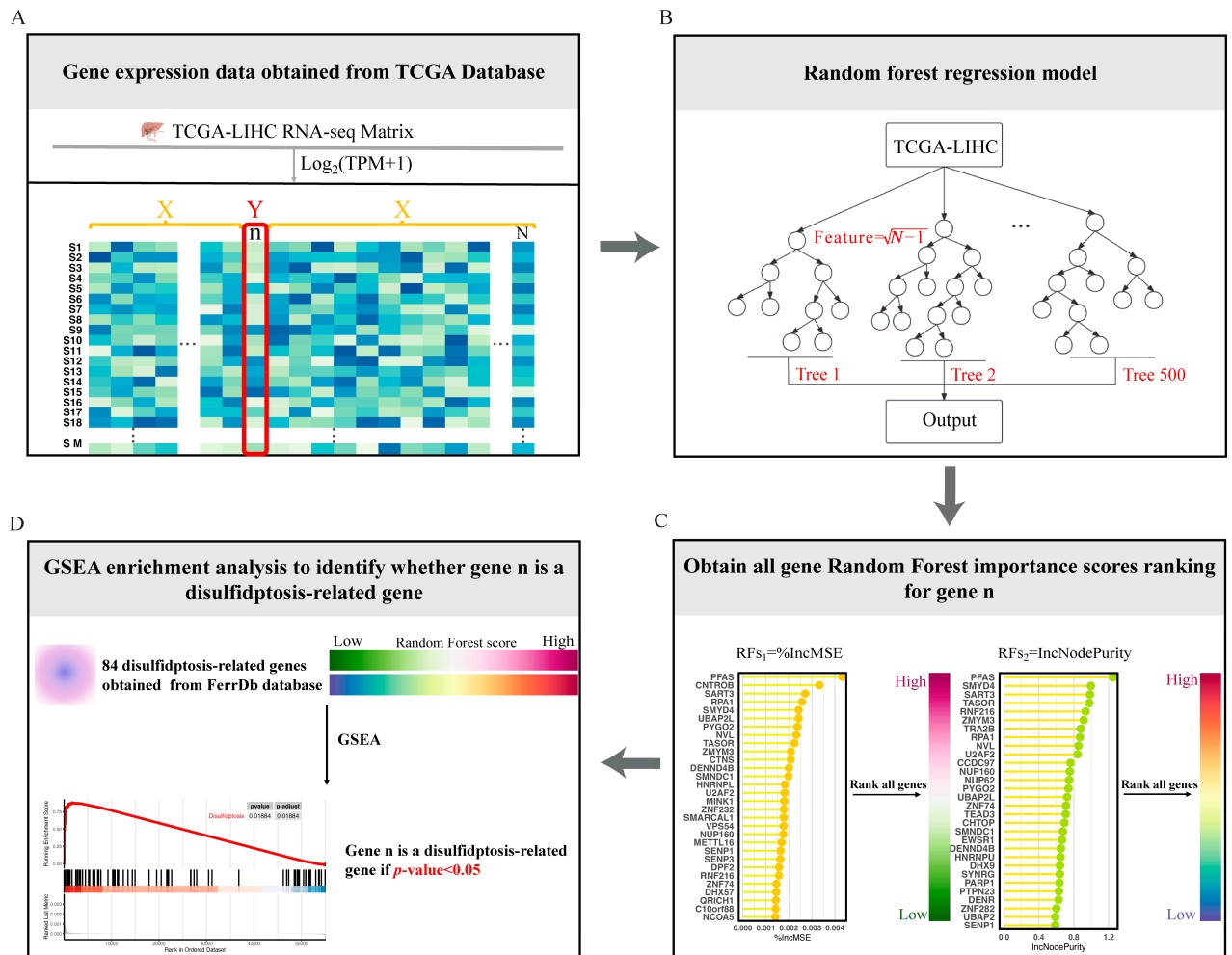

**Figure 2.** Overview of the four steps of the RF-GSEA for identifying disulfidptosis-related genes. (**A**) Obtain RNA-seq data for HCC from the TCGA database. (**B**) Construct a random forest regression model. (**C**) Obtain random forest scores based on the variable importance of genes. (**D**) GSEA identifies candidate disulfidptosis-related gene.

### 2.3. Consensus Clustering Analysis

Based on the expression of disulfidptosis-related genes, we performed consensus cluster analysis to identify different disulfidptosis-related clusters. By using the "ConsensusClusterPlus" package, we chose the K-means algorithm as the clustering algorithm and Euclidean distance as the distance calculation method. We sampled 80% of the samples 1000 times in each algorithm. The optimal number of clusters is determined based on the area under the cumulative distribution function (CDF) curve and the Delta Area Plot. The "Limma" package was used to identify differentially expressed genes (DEGs) between disulfidptosis-related clusters with a |log2FC| > 1 and p-adj < 0.05.

### 2.4. Functional Enrichment Analysis

We employed the "clusterProfiler" package for conducting Gene Ontology (GO) (including Biological Process, Molecular Function, and Cellular Component) functional enrichment analysis. This was performed to elucidate the biological functions of differentially expressed genes between distinct clusters associated with disulfide death. Functional enrichments with a false discovery rate (FDR) less than 0.05, corrected using the Benjamini–Hochberg (BH) method, were considered significantly enriched.

### 2.5. Identify a Disulfidptosis-Related Gene Signature

Lasso Cox regression analysis was employed to identify disulfidptosis-related genes associated with the overall survival (OS) of HCC patients. Based on these genes, we constructed a gene set for Gene Set Variation Analysis (GSVA) to obtain disulfidptosis-related scores. Subsequently, we conducted a time-dependent Receiver Operating Characteristic (ROC) analysis for the disulfidptosis-related score (DR score). Using the 12-month ROC curve, the optimal cutoff was selected according to Youden's index, and the samples were categorized into high- and low-DR score groups.

### 2.6. The Protein Expression Levels of Disulfidptosis-Related Signature Genes

To further validate the protein expression levels of disulfidptosis-related signature genes in HCC tumors and normal tissues, immunohistochemistry (IHC) data can be downloaded from the Human Protein Atlas database (HPA, http://www.Proteinatlas.org (accessed on 1 October 2023)). HPA provides a variety of IHC results for numerous proteins based on proteomics in both tumor and normal tissues.

### 2.7. Evaluation of the Prognostic Value of Disulfidptosis-Related Signature

Kaplan–Meier analysis was employed to assess the prognostic value of the disulfidptosis-related signature in three datasets (TCGA-LIHC, GSE116174, and LIRI-JP). We incorporated the disulfidptosis-related score along with other clinical factors (including Age, Gender, TNM_Stage, Grade, Alcohol, Smoking, Invasion, and HBV) into Cox regression analysis to evaluate whether it serves as an independent prognostic indicator for HCC.

### 2.8. Assessment of the Tumor Immune Microenvironment

We obtained a set of 28 immune cell gene signatures from the study by Barbie et al. [25] Subsequently, we used ssGSEA analysis to investigate whether there were differences in the immune infiltration levels among different DR score groups. In order for the anti-tumor immune response to effectively eliminate tumor cells, the organism must initiate a series of sequential events and allow for their iterative expansion. This process is referred to as the Cancer–Immunity Cycle. In cancer patients, the Cancer–Immunity Cycle is not always optimal, and various factors can impede its progression, leading to the immune evasion of tumor cells [26]. We analyzed the Cancer Immunity Cycle in different DR groups using the Tracking Tumor Immunophenotype (TIP) database (http://biocc.hrbmu.edu.cn/TIP/index.jsp (accessed on 4 October 2023)) to examine the key factors contributing to differential prognoses between the two groups.

### 2.9. Prediction of Immunotherapy Response

We assessed the correlation between the expression levels of four representative immune checkpoint molecules (LAG3, PDCD1, CTLA4, and CD274) and the DR score. The Tumor Immune Dysfunction and Exclusion (TIDE) algorithm is an effective tool for predicting immunotherapy responses. Using the TIDE algorithm, we compared the TIDE score, Dysfunction score, and Exclusion score between high- and low-DR score groups. Finally, we employed a real-world cohort (IMvigor210) undergoing anti-PD-L1 immunotherapy to assess the potential predictive value of the disulfidptosis-related signature in immunotherapy response.

### 2.10. Drug Sensitivity Analysis

In addition to immunotherapy, the primary treatment modality for HCC remains chemotherapy. We downloaded the training dataset for the ridge regression model from the Genomics of Drug Sensitivity in Cancer (GDSC) database (https://www.cancerrxgene.org/ ((accessed on 7 October 2023))) and calculated drug IC50 values for HCC samples using the "oncoPredict" package. By comparing the IC50 values between the high- and low-DR score groups, we assessed the potential value of the disulfidptosis-related signature in predicting chemotherapeutic drug sensitivity.

### 2.11. Statistical Analysis

Survival analysis was performed using the Log-rank test; comparisons of two groups with continuous variables were conducted using Wilcoxon tests; comparisons of two groups with categorical variables were assessed using the Chi-square test; and correlations between two groups with continuous variables were evaluated using Spearman correlation analysis. All tests were two-tailed, with statistical significance defined as $p < 0.05$. All analyses were conducted using R 4.2.1 software.

## 3. Results

### 3.1. Identification of Disulfidptosis-Related Genes Using RF-GSEA

We utilized RF-GSEA to identify two candidate disulfidptosis gene sets based on %IncMSE (3856 genes) and IncNodePurity (5490 genes) (Figure 3A,B, Tables S2–S4). The 84 disulfidptosis-related genes as prior information genes were identified 31 (1 validated gene, 4 screened genes, and 26 deduced genes) in the %IncMSE gene set and 39 (4 validated gene, 6 screened genes, and 29 deduced genes) in the IncNodePurity gene set (Figure 3C). Based on the Delta Area Plot and CDF curve (Figure S1A,B), we determined the optimal number of clusters to be two (Figure 3D, Table S5). In comparison to Cluster 2, Cluster 1 exhibited poorer prognoses in terms of OS, PFI, DFI, and DSS (Figure S1C–F). "limma" analysis identified a total of 1412 DEGs between the two clusters, with 1302 genes upregulated and 110 genes downregulated (Figure 3E, Table S6). Theoretically, in the random forest model, the use of %IncMSE as variable importance scores can lead to negative values. GSEA might successfully enrich due to these negative values. This situation indicates that gene *n* is highly unlikely to be associated with disulfidptosis. Therefore, we further divided the %IncMSE gene set into two subsets, namely, %IncMSE-Positive (3436 genes) and %IncMSE-Negative (420 genes). The %IncMSE-Positive gene set includes genes enriched in the positive direction in GSEA, indicating a high likelihood that these genes are associated with disulfidptosis. The %IncMSE-Negative gene set, on the other hand, includes genes enriched in the negative direction in GSEA, indicating that these genes are unlikely to be associated with disulfide death. As anticipated, in the Venn diagram, the %IncMSE-Negative gene set has only 20 genes overlap with the IncNodePurity gene set and with no gene overlap observed with the DEGs. This demonstrates the robustness of the RF-GSEA method. Due to the segregation of the %IncMSE-Negative gene set from the %IncMSE gene set, the four conditions do not overlap. Subsequent analysis only requires the genes overlapping among the %IncMSE-Positive gene set, the IncNodePurity gene set, and the DEGs (Figure 3F). In the end, we identified a total of 220 disulfidptosis-related genes (Table S7). Using Lasso Cox regression analysis, we ultimately identified a disulfidptosis-related signature comprising seven genes (GAS2L3, SPINDOC, CCT5, CCDC34, G6PD, MARCKSL1, and STC2) (Figure 4A, Table S8). The IHC results obtained from the HPA database revealed that the expression levels of all seven proteins are higher in HCC tumor tissues compared to normal liver tissues (Figure 5).

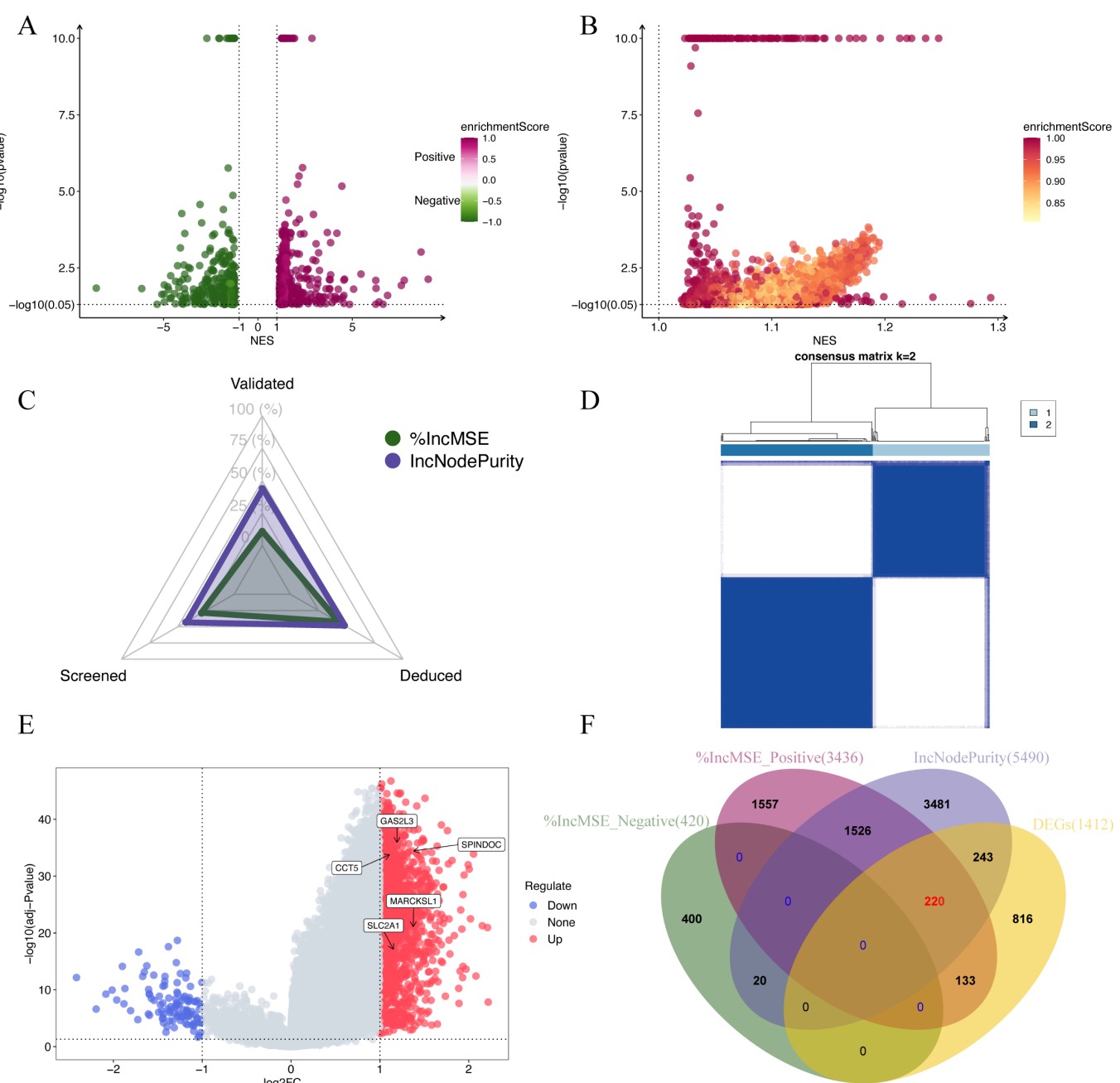

**Figure 3.** Identification of disulfidptosis-related genes. (**A**,**B**) Candidate disulfidptosis-related genes identified using the RF-GSEA pipeline (based on %IncMSE and IncNodePurity). Genes with *p*-values smaller than $1 \times 10^{-10}$ were plotted with a uniform *p*-value of $1 \times 10^{-10}$ for visualization purposes. (**C**) The status of the 84 genes identified in the %IncMSE gene set and the IncNodePurity gene set. (**D**) Consensus matrix of k = 2 based on the disulfidptosis gene set obtained from FerrDb dataset. (**E**) A volcano plot illustrating the Differentially Expressed Genes (DEGs) of "limma" analysis. (**F**) A Venn diagram depicting the final selection of 220 disulfidptosis-related genes.

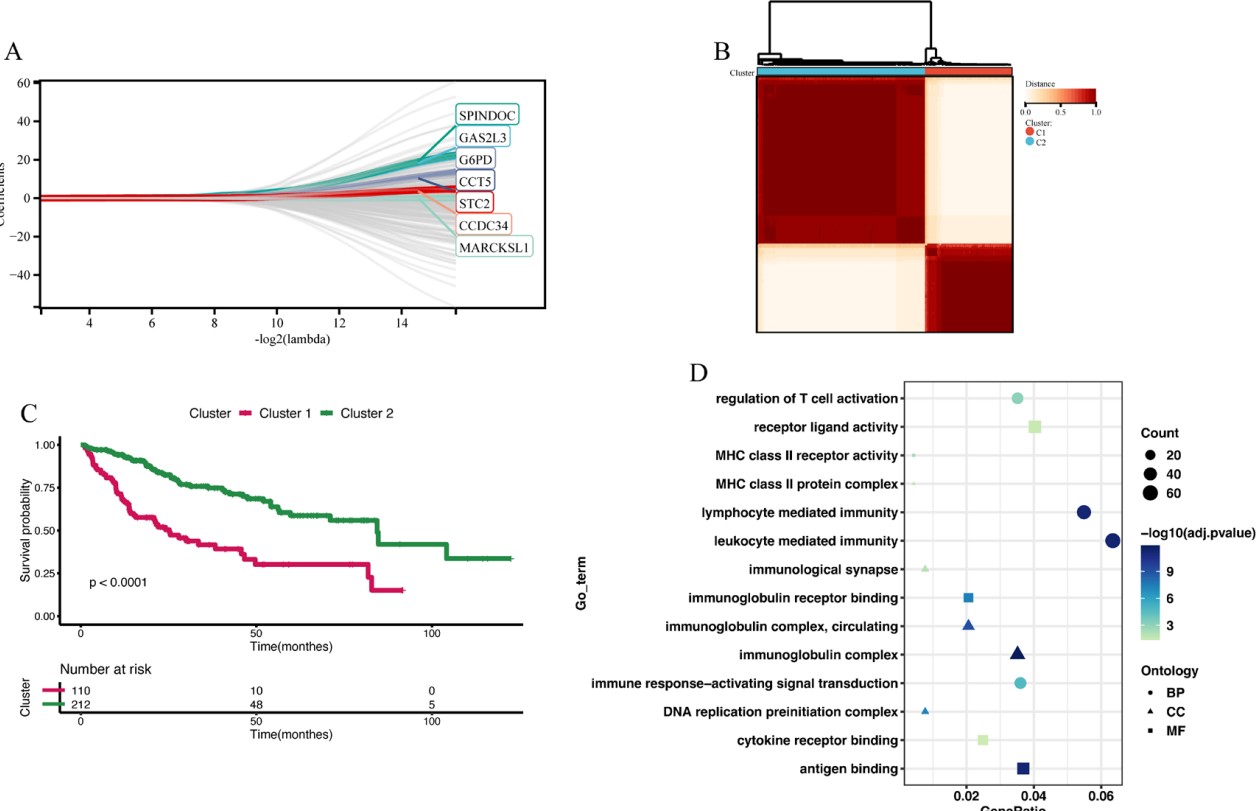

**Figure 4.** Constructing 2 disulfidptosis-related clusters. (**A**) Lasso Cox regression analysis identified a disulfidptosis-related signature comprising seven genes. (**B**) Consensus matrix of 2 disulfidptosis-related clusters. (**C**) The Kaplan–Meier analysis of OS between Cluster 1 and Cluster 2. (**D**) GO enrichment analysis of DEGs between Cluster 1 and Cluster 2.

### 3.2. Pan-Cancer Expression Pattern, Prognostic Significance, and Immunological Correlation of SLC7A11

Before applying the RF-GSEA method to HCC, it is essential to determine whether disulfidptosis plays a significant role in HCC. Researchers who have discovered disulfidptosis believe that SLC7A11 plays a pivotal role in this process. We conducted a comprehensive pan-cancer analysis of SLC7A11 to unveil the potential involvement of disulfidptosis in cancer. We observed that in nearly all types of cancers, the expression level of SLC7A11 in tumor tissues is significantly higher than that in normal tissues (Figure S2). In HCC, the expression of SLC7A11 is also higher in patients at later stages (Stage III + IV, Grade G3 + G4) compared to those at earlier stages (Stage I + II, Grade G1 + G2). Furthermore, single-cell sequencing analysis revealed a significant expression of SLC7A11 in malignant cells of HCC (Figure S3). In many cancers, high expression of SLC7A11 is indicative of a poorer prognosis. In HCC, SLC7A11 serves as an independent prognostic factor, surpassing other clinical characteristics (Figure S4). ssGSEA was used to assess the relationship between SLC7A11 and 28 types of immune cells. We observed significant variations in the relationship between SLC7A11 and immune cell infiltration levels across different tumors. In PCPG, KIRP, and KIRC, SLC7A11 exhibited a positive correlation with the infiltration levels of nearly all immune cells. In contrast, in THCA, LUSC, and LUAD, SLC7A11 showed a negative correlation with the infiltration levels of nearly all immune cells. In our study of the correlation between the Cancer–Immunity Cycle and SLC7A11, we also observed similar phenomena (Figure S5). These findings suggest that disulfidptosis may play a significant role in various cancers, including HCC.

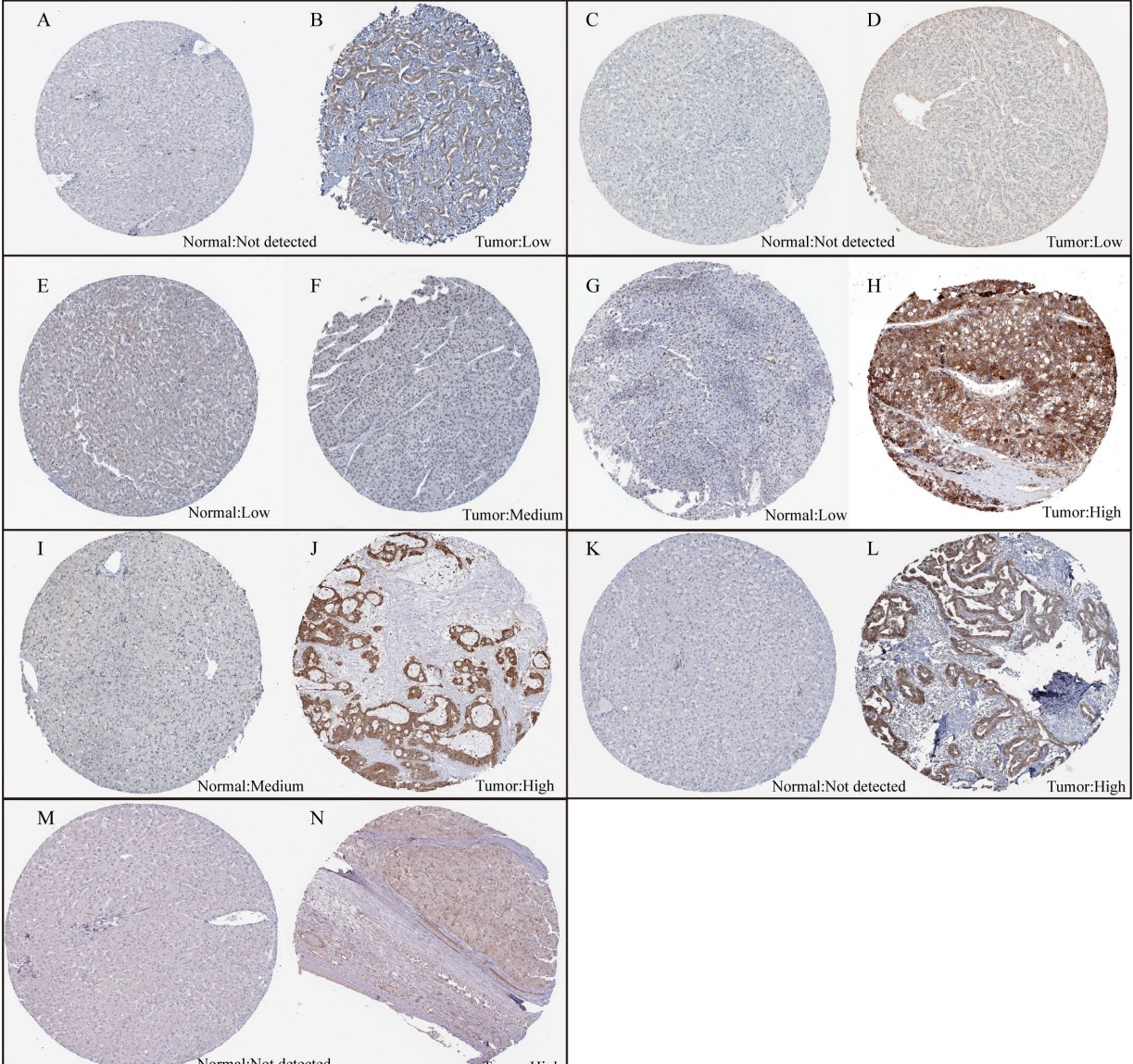

**Figure 5.** The expression levels of the seven proteins between normal liver tissue and HCC tumor tissue. (**A,B**) The expression levels of SPINDOC between normal liver tissue and HCC tumor tissue. (**C,D**) The expression levels of CCT5 between normal liver tissue and HCC tumor tissue. (**E,F**) The expression levels of CCDC34 between normal liver tissue and HCC tumor tissue. (**G,H**) The expression levels of G6PD between normal liver tissue and HCC tumor tissue. (**I,J**) The expression levels of MARCKSL1 between normal liver tissue and HCC tumor tissue. (**K,L**) The expression levels of STC2 between normal liver tissue and HCC tumor tissue. (**M,N**) The expression levels of GAS2L3 between normal liver tissue and HCC tumor tissue.

### 3.3. Construction of Disulfidptosis-Related Clusters

Consensus clustering analysis was utilized to construct two disulfidptosis-related clusters (Figure 4B and Figures S6A–D, Table S9). Compared to Cluster 2, Cluster 1 exhibited poorer prognoses in terms of OS (Figure 4C), PFI, DFI, and DSS (Figure S6E–G). Between Cluster 1 and Cluster 2, the "limma" analysis identified a total of 1264 DEGs, comprising 263 downregulated genes and 1001 upregulated genes (Table S10). These genes were subjected to GO enrichment analysis, revealing their primary involvement in immune regulatory functions. These functions included immune response-activating signal transduction, lymphocyte-mediated immunity, immunological synapse formation, and immunoglobulin receptor binding (Figure 4D). Furthermore, we explored the tumor

microenvironment differences between the two clusters. In Cluster 1, the infiltration levels of immune cells such as activated CD4 T cells, activated dendritic cells, central memory CD4 T cells, central memory CD8 T cells, and effector memory CD4 T cells were higher than in Cluster 2. However, the infiltration level of eosinophils was lower in Cluster 1 compared to Cluster 2 (Figure S7A). Comparing the Cancer–Immunity Cycle status between the two clusters, we observed that while Cluster 1 outperformed Cluster 2 in the earlier steps (Step 1: release of cancer cell antigens; Step 2: neutrophil recruiting, Th17 cell recruiting, and Treg cell recruiting), it exhibited poorer performance in the later steps (Step 5: infiltration of immune cells into tumors; Step 6: recognition of cancer cells by T cells; and Step 7: killing of cancer cells) compared to Cluster 2 (Figure S7B). The poorer performance in the later steps of the Cancer–Immunity Cycle explains why Cluster 1 has a worse prognosis compared to Cluster 2.

### 3.4. Identification and Validation of Disulfidptosis-Related Score

The final set of seven genes was utilized as the gene set, and GSVA was employed to identify the DR score. Time-dependent ROC analysis was employed to assess the prognostic value of the DR score in three cohorts (TCGA-LIHC, GSE116174, and LIRI-JP). We observed that the DR score exhibited significant predictive value in all three cohorts (Figure S8). Based on the 12-month time-dependent ROC curve in TCGA-LIHC, we chose a cutoff score of 0.381, which corresponded to the maximum Youden index, to classify all cohorts into high- and low-DR score groups (Figure S8A). In all three cohorts, we consistently observed that the OS prognosis was worse in the high-DR score group compared to the low-DR score group (Figure 6A,E,I). Furthermore, the distribution plots, survival status plots, and gene expression heatmaps provided additional evidence of the robust discriminative capacity of the disulfidptosis-related signature for HCC patients (Figure 6B–D,F–H,J,K). The results of univariate Cox regression analysis indicated that in the TCGA-LIHC cohort, the disulfidptosis-related signature and stage were risk factors for OS (Figure 7A). In the GSE116174 cohort, the disulfidptosis-related signature and invasion were risk factors for OS (Figure 7C). In the LIRI-JP cohort, the disulfidptosis-related signature and stage were risk factors for OS (Figure 7E). Subsequently, we conducted multivariate Cox regression analysis and found that in all three cohorts, the disulfidptosis-related signature remained an independent risk factor for OS (Figure 7B,D,F). These results indicate that the disulfidptosis-related signature holds significant prognostic value in HCC patients.

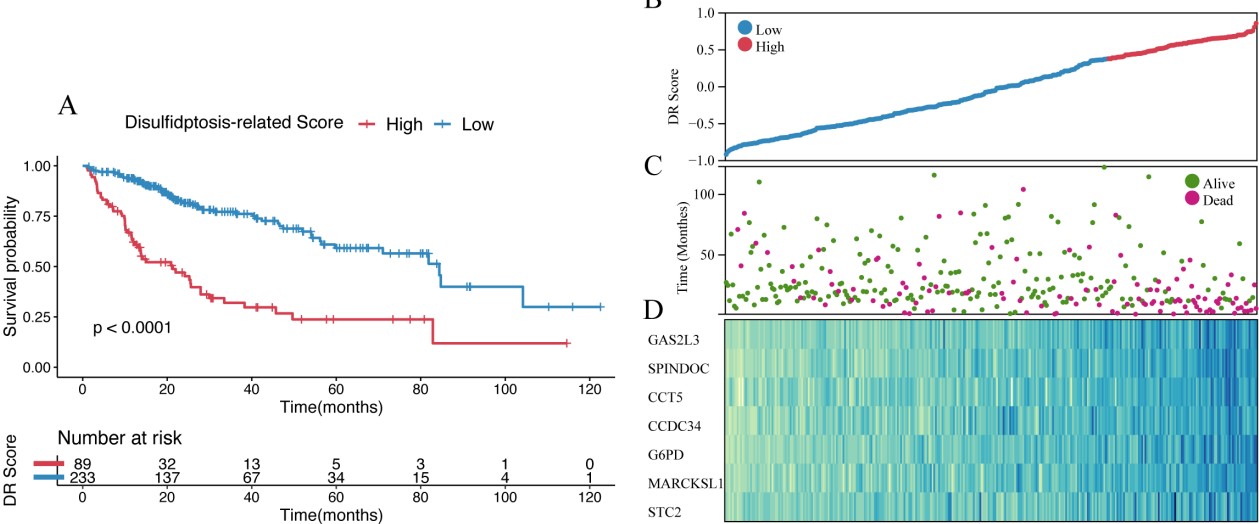

**Figure 6.** *Cont.*

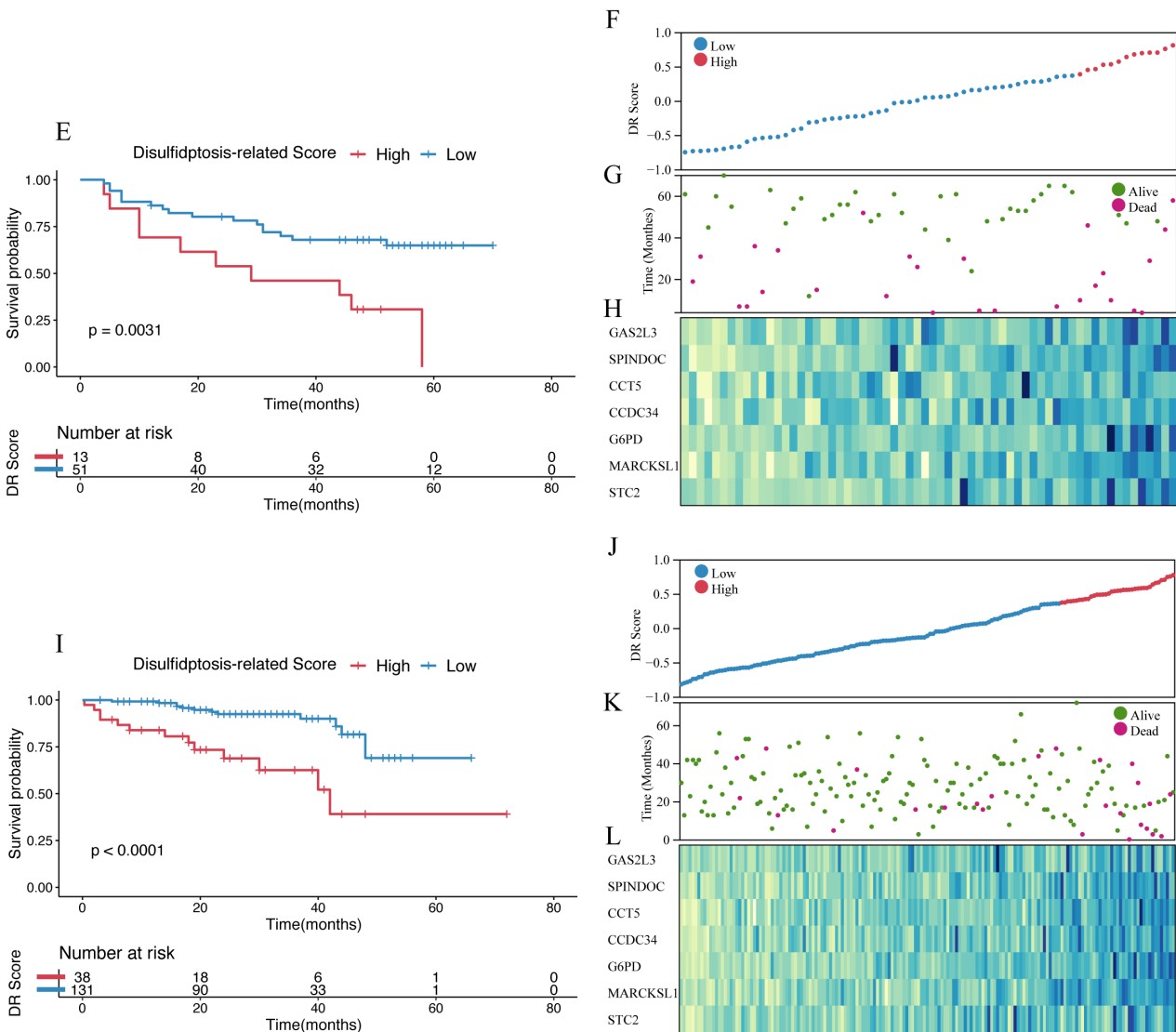

**Figure 6.** Identification and validation of disulfidptosis-related score. (**A**) The Kaplan–Meier survival analysis of OS between high- and low-score groups in the TCGA-LIHC cohort. (**B**) The distribution of the TCGA-LIHC cohort based on DR score. (**C**) The survival status of the TCGA-LIHC cohort based on DR score. (**D**) The heatmap of 7 disulfidptosis-related genes between high- and low-score groups in the TCGA-LIHC cohort. (**E**) The Kaplan–Meier survival analysis of OS between high- and low-score groups in the GSE116174 cohort. (**F**) The distribution of the GSE116174 cohort based on DR score. (**G**) The survival status of the GSE116174 cohort based on DR score. (**H**) The heatmap of 7 disulfidptosis-related genes between high- and low-score groups in the GSE116174 cohort. (**I**) The Kaplan–Meier survival analysis of OS between high- and low-score groups in the LIRI-JP cohort. (**J**) The distribution of the LIRI-JP cohort based on DR score. (**K**) The survival status of the LIRU-JP cohort based on DR score. (**L**) The heatmap of 7 disulfidptosis-related genes between the high- and low-score groups in the LITI-JP cohort.

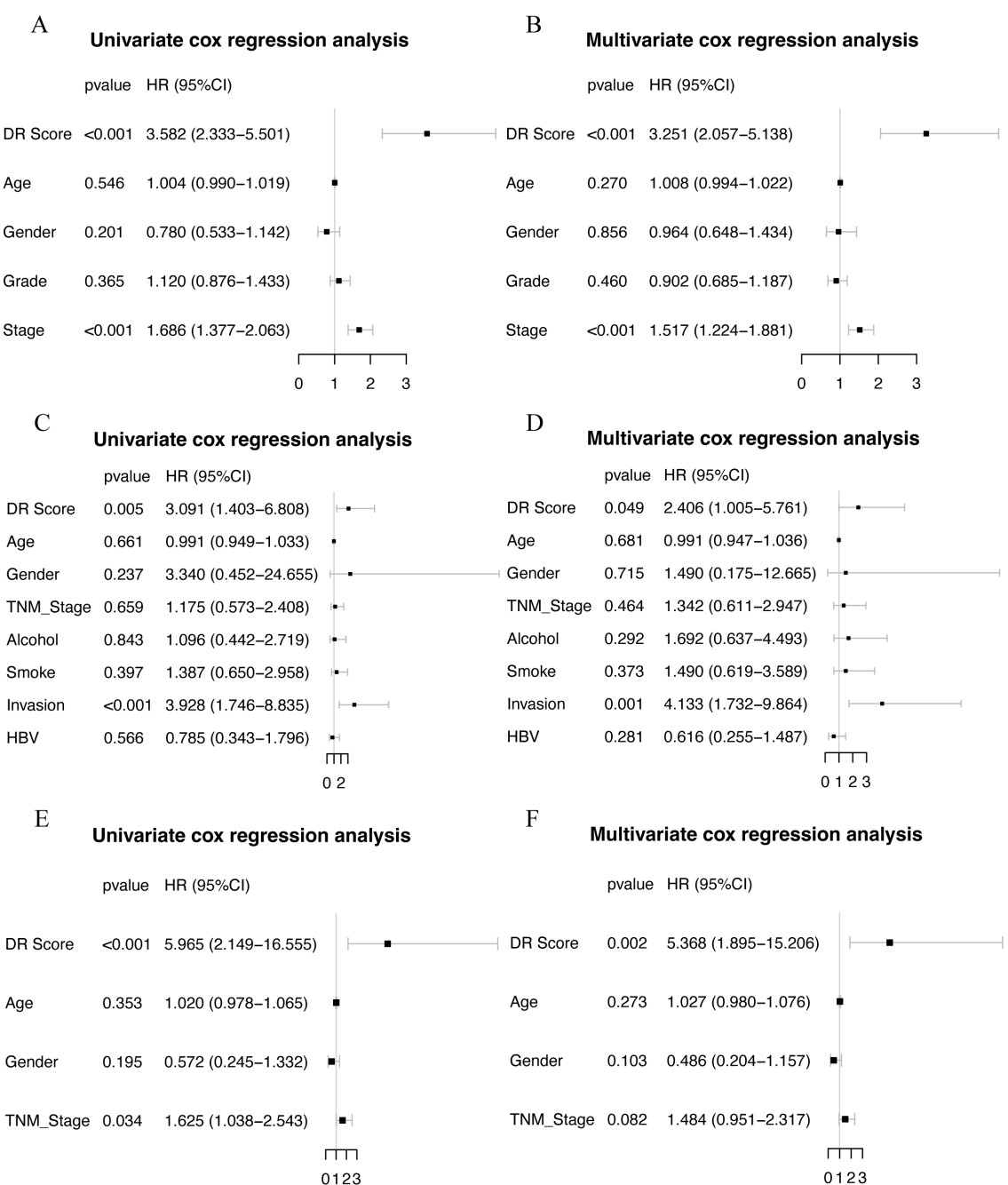

**Figure 7.** Univariate and multivariate COX regression in three cohorts. (**A**,**B**) Univariate and multivariate COX regression in TCGA-LIHC cohort. (**C**,**D**) Univariate and multivariate COX regression in GSE116174 cohort. (**E**,**F**) Univariate and multivariate COX regression in LIRI-JP cohort.

### 3.5. The Relationship with Tumor Microenvironment

We further investigated the role of the disulfidptosis-related signature in the tumor microenvironment of HCC. ssGSEA revealed that among the 28 immune-related cell types, the high-DR score group exhibited higher levels of infiltration for activated CD4 T cells, activated dendritic cells, central memory CD4 T cells, effector memory CD4 T cells, regulatory T cells, and type 2 T helper cells compared to the low-DR score group. Conversely, the high-DR score group displayed lower infiltration levels for activated CD8 T cells, CD56 bright natural killer cells, and eosinophils compared to the low-DR score group (Figure 8A). Comparing the Cancer–Immunity Cycle status between the two groups, we observed that the high-DR score group exhibited higher levels in Step 1 (release of cancer cell antigens),

Step 4 (neutrophil recruiting), Step 4 (Th17 cell recruiting), and Step 4 (MDSC recruiting compared to the low-DR score group). On the contrary, the high-DR score group displayed lower levels in Step 3 (priming and activation), Step 4 (CD4 T cell recruiting), Step 5 (infiltration of immune cells into tumors), and Step 7 (killing of cancer cells) compared to the low-DR score group (Figure 8B).

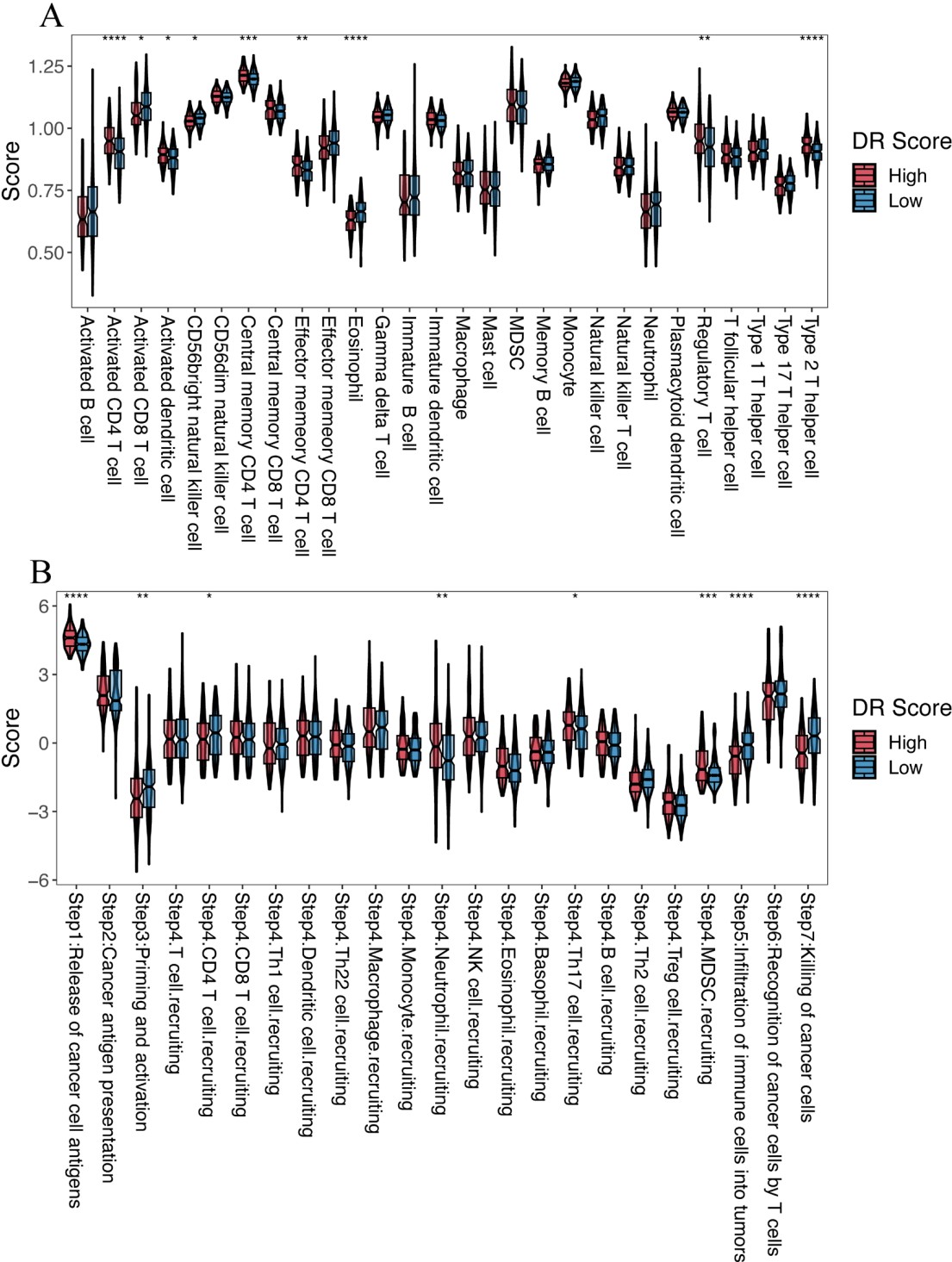

**Figure 8.** The relationship between the disulfidptosis-related signature and the tumor microenvironment. (**A**) Comparison of the infiltration levels of 28 immune cell types between the high- and low-DR score groups. (**B**) Cancer–Immunity Cycle status between the high- and low-DR score groups. Kruskal–Wallis test; * $p < 0.05$, ** $p < 0.01$, *** $p < 0.001$ and **** $p < 0.0001$.

### 3.6. Prediction of the Efficacy of Immunotherapy and Targeted Anti-Tumor Agents

We investigated the relationship between four representative immune checkpoint genes (LAG3, PDCD1, CTLA4, and CD274) and the disulfidptosis-related signature. We found a significant positive correlation between disulfidptosis-related signatures and their expression level (Figure 9A–D). We also explored the correlation between the disulfidptosis-related signature and 91 immunomodulators [27], and we were surprised to find that the disulfidptosis-related signature exhibited significant correlations with the expression levels of almost all of these genes (Table S11). Furthermore, using the TIDE algorithm, we found that the high-DR score group had significantly lower TIDE, Dysfunction, and Exclusion scores compared to the low-DR score group (Figure 9E–G). A higher TIDE score indicates a higher likelihood of immune evasion and a lower likelihood of benefiting from immunotherapy. The relationship between the DR score and immune checkpoint, as well as TIDE scores, indicates that the higher the DR score, the more likely to benefit from immunotherapy. In our previous pan-cancer analysis of SLC7A11, we found that the expression pattern, prognostic value, and its correlation with immune regulation genes, tumor microenvironment, and RNA modification genes in BLCA were highly similar to those in HCC. Based on this discovery, we believe that the role of disulfidptosis in BLCA is similar to its role in HCC. Due to the lack of authoritative real-world immunotherapy cohorts for HCC, we chose the reputable BLCA immunotherapy cohort IMvigor210 to further investigated the relationship between the disulfidptosis-related signature and the response to immunotherapy. Consistent with our previous findings, the higher the disulfidptosis-related score, the more likely to benefit from immunotherapy (Figure 9H,I). The diversity of HCC treatment highlights that we should prioritize more than just immunotherapy, given that a wide array of targeted anti-tumor agents continues to play a crucial role in HCC treatment. We compared the differences in IC50 values for eight chemotherapy drugs, including the first-line treatment drug Sorafenib, between different DR score groups. We were surprised to find that the IC50 values for these eight chemotherapy drugs were all lower in the high-DR score group compared to the low-DR score group (Figure 10A–H). This indicates that, relative to the low group, the high group exhibits higher sensitivity to these eight drugs.

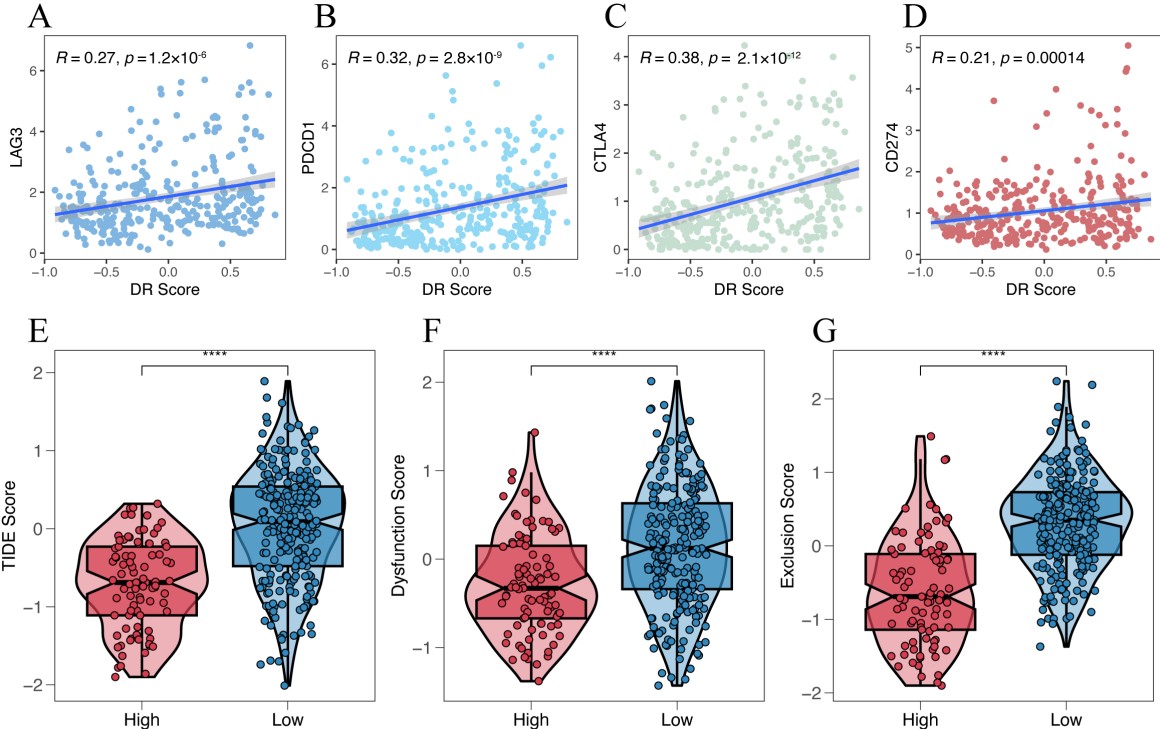

**Figure 9.** *Cont.*

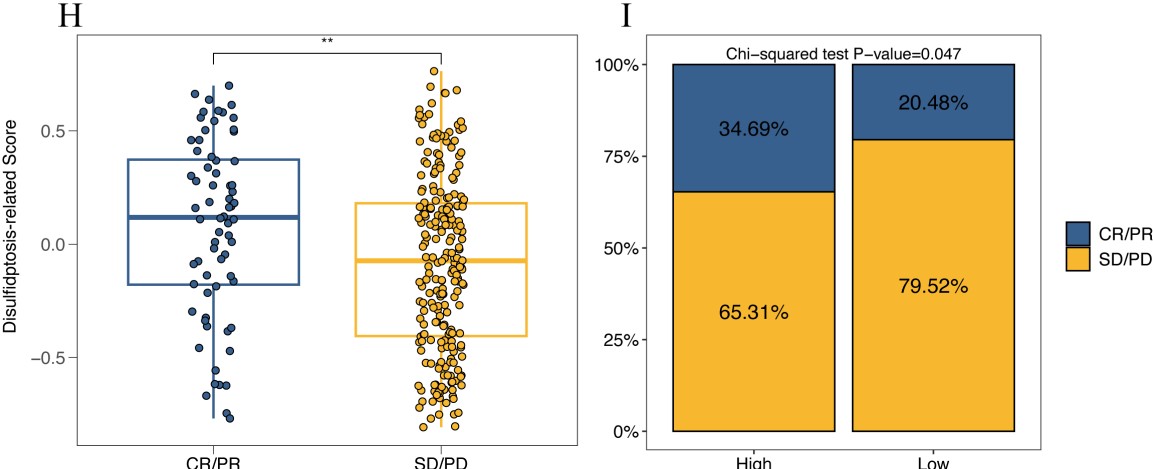

**Figure 9.** Prediction of the efficacy of immunotherapy. (**A–D**) The relationship between disulfidptosis-related signature with 4 immune checkpoints (LAG3, PDCD1, CTLA4 and CD274). (**E–G**) The relationship between disulfidptosis-related signature with TIDE, Dysfunction and Exclusion score. (**H**) Comparison of disulfidptosis-related scores for different immunotherapy outcomes. (**I**) The proportion of different immunotherapy outcomes between high- and low-DR score groups. Kruskal–Wallis test; ** $p < 0.01$ and **** $p < 0.0001$.

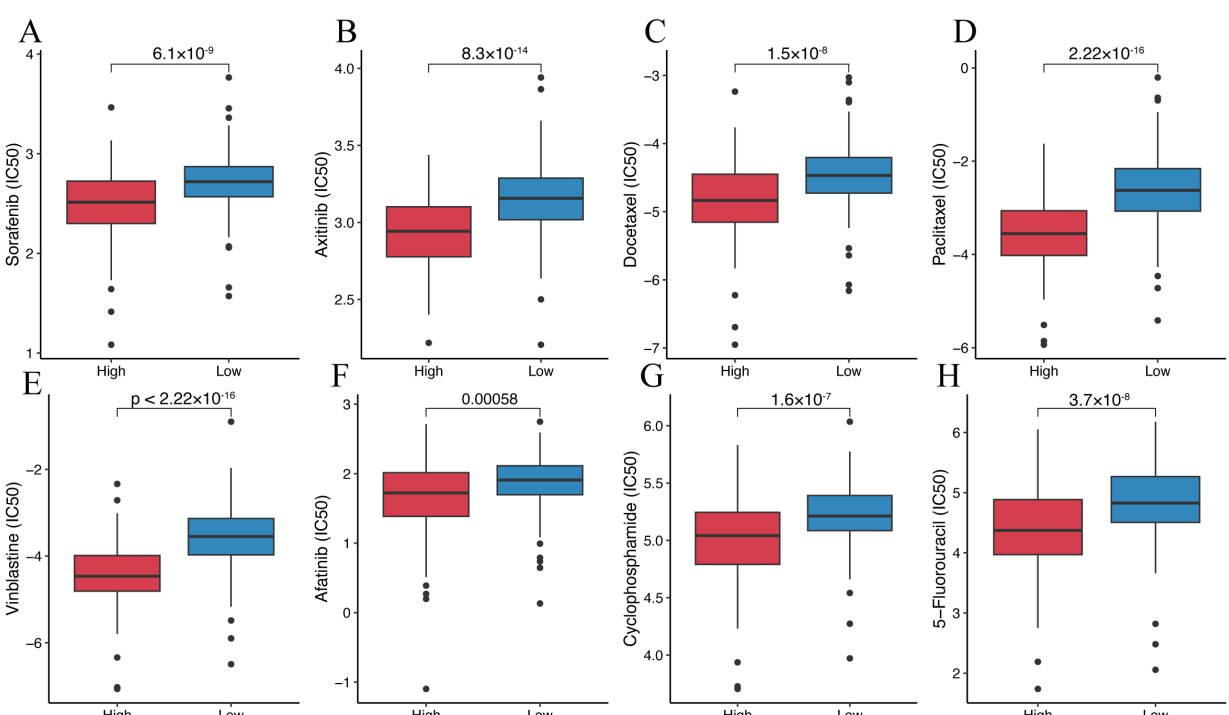

**Figure 10.** Prediction of the efficacy of targeted anti-tumor agents. (**A**) Comparison of Sorafenib IC50 between the high- and low-DR score groups. (**B**) Comparison of Axitinib IC50 between the high- and low-DR score groups. (**C**) Comparison of Docetaxel IC50 between the high- and low-DR score groups. (**D**) Comparison of Paclitaxel IC50 between the high- and low-DR score groups. (**E**) Comparison of Vinblastine IC50 between the high- and low-DR score groups. (**F**) Comparison of Afatinib IC50 between the high- and low-DR score groups. (**G**) Comparison of Cyclophosphamide IC50 between the high- and low-DR score groups. (**H**) Comparison of 5-Fluorouracil IC50 between the high- and low-DR score groups.

## 4. Discussion

In this study, we developed a method that combines machine learning with enrichment analysis based on gene regulatory networks and graph theory to identify potential disulfidptosis-related genes. This method demonstrates a robust capability for the identification of disulfidptosis-related genes, thereby reducing the number of candidate genes to be experimentally validated. As expected, there was minimal overlap of only 20 genes between the %IncMSE-Negative gene set and the IncNodePurity gene set, with no overlap observed with the EDGs. This underscores the rationality and robustness of our methodology. The RF-GSEA method identified a total of 220 disulfidptosis-related genes. These genes are predominantly associated with actin and cytoskeletal proteins, as well as the synthesis of glucose metabolites, suggesting their potential involvement in the regulation of disulfidptosis [28–30]. The biological functions of these genes further validate the robustness of the RF-GSEA pipeline. As a newly discovered mode of programmed cell death, the value of disulfidptosis in HCC has not yet been fully elucidated [31]. To validate the practical utility of the RF-GSEA method, we applied it to investigate the role of disulfidptosis in HCC.

We conducted a comprehensive pan-cancer analysis of SLC7A11, a gene that plays a significant role in disulfidptosis. We found that, although the expression levels of SLC7A11 in tumor tissues were elevated compared to normal tissues in almost all types of cancer, its relationship with the tumor microenvironment varied greatly across different cancer types. This phenomenon may be attributed to two potential reasons. Firstly, tumors are complex systems, and the tumor microenvironment may exhibit significant variations depending on the organ where the cancer originates [32]. Secondly, SLC7A11's role in programmed cell death serves as a double-edged sword [2–4]. It can both inhibit ferroptosis and initiate disulfidptosis. In different tumors, its primary function may undergo distinct alterations. This discovery suggests that disulfidptosis indeed plays a significant role in cancer, but its role may vary across different cancer types. Subsequently, lasso Cox analysis ultimately screened seven genes for constructing the disulfidptosis-related signature. Two clusters were identified based on these seven genes. Compared to Cluster 2, Cluster 1 exhibited a worse prognosis and higher infiltration of MDSCs and Tregs. Previous research has already indicated that high infiltration of MDSCs and Tregs leads to a poor prognosis in HCC, aligning with our study findings [33,34]. Furthermore, Cluster 1 exhibited higher scores in the neutrophil, Treg cell, and MDSC recruiting steps of the cancer immune cycle compared to Cluster 2. In addition to MDSCs and Tregs, neutrophils have also been confirmed to play a significant role in immune suppression in HCC [34]. In the final three steps of killing tumor cells, Cluster 1 performed less effectively than Cluster 2. This may be a direct contributing factor to the poorer prognosis observed in Cluster 1 [26]. The functional enrichment analysis of the EDGs between the two clusters revealed their primary enrichment in immune responses. This suggests a close association between the disulfidptosis-related signature and tumor immunity.

We developed a disulfidptosis-related scoring system using GSVA and categorized HCC samples into high- and low-DR score groups based on a cutoff. In both TCGA-LIHC and two external validation cohorts (GSE116174 and LIRI-JP), we consistently observed a poorer prognosis in the high-DR score group compared to the low-DR score group. The results of the Cox regression analysis further underscore that the disulfidptosis-related signature is an independent risk factor for HCC prognosis. The high-DR score group exhibited lower infiltration levels of activated CD8 T cells, CD56 bright natural killer cells, and eosinophils compared to the low-DR score group. In cancer, activated CD8 T cells play a primary role in killing cancer cells [35]. The low level of activated CD8 T cells in the high-DR score group suggests an inadequate ability to eliminate cancer cells. The primary function of CD56 bright natural killer cells is to initiate innate immune responses against cancer cells [36]. Abundant infiltration of CD56 bright natural killer cells is indicative of a better prognosis in HCC and is positively correlated with the apoptosis of tumor cells in HCC [37]. Furthermore, previous research has also indicated a positive correlation

between eosinophils and the prognosis of HCC [38,39]. In contrast, the high-DR score group exhibited higher levels of infiltration of activated CD4 T cells, activated dendritic cells, central memory CD4 T cells, effector memory CD4 T cells, regulatory T cells, and type 2 T helper cells compared to the low-DR score group. Current research has found that CD4 T cells can both promote and inhibit the generation of CD8+ cytotoxic T cells. This complex role has led to an ongoing debate regarding the impact of CD4 T cells on tumor prognosis [40]. In our study, many CD4 T cells in the high-DR score group, associated with poorer survival, exhibited high levels of infiltration. This observation may suggest that in HCC, CD4 T cells play an immunosuppressive role. In the cancer immune cycle, the high-DR score group displayed higher levels of release of cancer cell antigens, neutrophil recruiting, Th17 cell recruiting, and MDSCs recruiting compared to the low-DR score group. The higher level of release of cancer cell antigens in the high-DR score group suggests an increased production of new antigens due to tumor occurrence, which are subsequently released and captured by dendritic cells for processing. However, this represents only the initial step in the entire process of killing tumor cells. In the subsequent steps, the high-DR score group appears to transition toward immune suppression, ultimately leading to their unfavorable prognosis [26]. The relevance of Th17 cells in promoting autoimmunity, carcinogenesis, and anti-tumor immunity has been established [41]. Although the high-DR score group recruited more Th17 cells, their ability to function effectively requires infiltration into the tumor tissue. However, in the ssGSEA analysis, we did not observe a higher level of Th17 cell infiltration in the high-DR score group. This suggests that the process of tumor-inhibitory Th17 cell infiltration into HCC tumor tissue is hindered, leading to the immunosuppressive state and poorer prognosis observed in the high-DR score group. Moreover, the excessive recruitment of cells such as neutrophils and MDSCs, which have been shown to play an immunosuppressive role in hepatocellular carcinoma [42] in the high-DR score group, also contributes to the unfavorable prognosis. In contrast, the high-DR score group exhibited lower levels of priming and activation, infiltration of immune cells into tumors, and killing of cancer cells compared to the low-DR score group. The poor performance of the high-DR score group in these steps directly explains its worse prognosis and immunosuppressive state.

Previous studies have already demonstrated that the high expression of these four immune checkpoint molecules (LAG3, PDCD1, CTLA4, and CD274) promotes T cell exhaustion, leading to immune evasion and an unfavorable prognosis in HCC [43,44]. The significant positive correlation between the DR score and these molecules further validates the value of the DR score for immune evasion and prognosis in HCC. In HCC, the higher the expression of immune checkpoint molecules, the more likely for patients to benefit from immunotherapy [45]. The TIDE score in the high-DR score group was lower than in the low-DR score group. A lower TIDE score indicates a higher likelihood of benefiting from immunotherapy [46]. Both the Spearman correlation analysis and the TIDE analysis suggest that the higher the DR score, the more likely patients are to benefit from immunotherapy. This discovery was further validated in real immunotherapy cohorts. Lastly, drug sensitivity analysis revealed that the IC50 values for eight chemotherapeutic drugs were all lower in the high-DR score group compared to the low-DR score group. This suggests that the high-DR score group exhibits greater sensitivity to these eight chemotherapeutic drugs. The combination of immunotherapy and targeted chemotherapy drugs has been approved for the treatment of HCC [47,48]. Understanding the potential response of patients to treatment aids in devising personalized treatment regimens. Our study suggests that immunotherapy and targeted chemotherapy drugs may have better efficacy in the high-DR score group.

*Strengths and Weaknesses*

This work presents the development of the RF-GSEA method, which possesses numerous advantageous features. The RF-GSEA method is an innovative approach for discovering genes linked to a certain biological process. The RF-GSEA method is valu-

able for the study of biological processes as well as gene function. The RF-GSEA method demonstrates a partial capacity to address the issue of mixed expression pattern genes. Genes with mixed expression patterns can be considered as outliers in machine learning. The random forest model employs multiple decision trees, each utilizing only a subset of features from the data; thus, outliers do not significantly impact the model. The RF-GSEA method also has some weaknesses that need to be addressed. The RF-GSEA method does not fully address the mixed expression pattern genes. In the current study, we had an insufficient sample size (TCGA-LIHC cohort, 322 samples) when using the RF-GSEA method. Despite our efforts to utilize existing data for model validation, it is crucial to note that the evidence provided by protein expression levels is limited. Further experimental validation is required to further validate our model.

## 5. Conclusions

The RF-GSEA method is a powerful tool for identifying potential disulfidptosis-related genes. In-depth research into the genes identified in this study can reveal the underlying mechanisms of disulfidptosis. To validate its practical utility, we applied it to HCC, leading to the development of a disulfidptosis signature. This signature accurately predicts HCC prognosis and unveils its underlying mechanisms. Furthermore, it can stratify individuals sensitive to immunotherapy and chemotherapy, offering guidance for personalized treatment strategies. Additionally, RF-GSEA is not only limited to the identification of potential disulfidptosis-related genes but also is equally applicable to other biological processes.

**Supplementary Materials:** The following supporting information can be downloaded at: https://www.mdpi.com/article/10.3390/cimb45120593/s1, Figure S1: Consensus clustering analysis based on 84 disulfidptosis-related genes; Figure S2: The expression level of SLC7A11 across different cancers; Figure S3: SLC7A11 expression pattern in HCC; Figure S4: The prognostic value of SLC7A11 in cancers; Figure S5: The correlation of SLC7A11 with the tumor microenvironment in various cancers; Figure S6: T Construction of disulfidptosis-related clusters; Figure S7: The relationship between the disulfidptosis-related clusters and the tumor microenvironment; Figure S8: Time-dependent ROC analysis of the DR score in three cohorts; Table S1: Disulfidptosis-related genes obtained from FerrDb database; Table S2: The %IncMSE_Negative gene set; Table S3: The %IncMSE_Positive gene set; Table S4: The IncNodePurity gene set; Table S5: Consensus Clustering analysis based on FerrDb gene set; Table S6: Limma differential expression analysis; Table S7: Disulfidptosis-related genes obtained from RF_GSEA method; Table S8: 7 disulfidptosis-related genes screened by lasso-cox; Table S9: Consensus Clustering analysis based on 7 disulfidptosis-related genes; Table S10: DEGs between disulfidptosis-related clusters; Table S11: Relationship between disulfidptosis-related score and Immunomodulators.

**Author Contributions:** Conceptualization, data curation, investigation, writing, review and editing, L.N. and Q.Y.; methodology, R.Y.; formal analysis, L.N., Q.Y., R.Y. and C.C.; validation, R.Y. and C.C.; visualization, L.N., Q.Y. and C.C; conceptualization, investigation, methodology, review and editing, B.P. All authors contributed to the article and approved the submitted version. All authors have read and agreed to the published version of the manuscript.

**Funding:** This research received no external funding.

**Informed Consent Statement:** Not applicable.

**Data Availability Statement:** The raw data for this study can be downloaded from the public databases mentioned in the article. Data generated from the study can be downloaded in the Supplementary Materials or https://github.com/little2b/the-RF-GSEA-Method (accessed on 22 November 2023).

**Acknowledgments:** The authors would like to thank the researchers who provided open access to the raw data.

**Conflicts of Interest:** The authors declare no conflict of interest.

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
