# Peer review of "Development of the RF-GSEA Method for Identifying Disulfidptosis-Related Genes and Application in Hepatocellular Carcinoma"

_cimb, doi:10.3390/cimb45120593_

Round 1
Reviewer 1 Report
Comments and Suggestions for Authors
The manuscript “Development of the RF-GSEA Method for Identifying Disulfidptosis-Related Genes and Application in Hepatocellular Carcinoma” discusses the use of the RF-GSEA method (Random Forest regression model and Gene Set Enrichment Analysis) for identifying potential genes associated with disulfidptosis and explores the relationship between the disulfidptosis-related signature and HCC (hepatocellular carcinoma) prognosis, tumor microenvironment, immunotherapy efficacy, and chemotherapy drug sensitivity.
The entire manuscript is well-written and adequately illustrated, with sufficient explanation of the corresponding procedures in the Materials and Method section and obtained data in the results section and accompanying supplementary materials. The following discussion section is appropriate, and the manuscript is complemented with a corresponding and up-to-date reference list.
I thank the authors for their efforts and the valuable data presented.
The acceptance of the manuscript in its current form is suggested.
Author Response
Thank you very much for your review and feedback. We are very pleased to hear that you felt the manuscript did not require revision. Thank you for recognizing and supporting our work, and we will continue to strive to ensure its quality and accuracy.
Reviewer 2 Report
Comments and Suggestions for Authors
How the authors determine the deduced genes group?
All the cohorts are using the same normalization method (log2(TPM+1) ?
How the mixed expression pattern genes are handled? Theoretically, we can't expect all genes will show uniformed expression pattern across patients. How the authors handled those list of genes?
Protein expression level section looks quiet weak and not convincing, it would be better provide more information and validation.
Figure 3A and B, remove outliers
Figure 3E, Set the FC >1.5 and < -1.5 and mark some key genes. Similarly Figure 3F doesn't provide much information, there is no overlap in all four conditions, better provide in some other form.
Figure 6, Survival plot, what is the cut point set ? spelling error in OS plot (months)
Fig.7 COX model is not clear enough, provide better figure (enlarged form)
Overall strength and weakness of this model should be discussed in discussion
minor
spelling error in line 18 mothed --> method
several places, lot of spelling and gramatical errors should be rectified.
Comments on the Quality of English LanguagePresent form is not very good, it has lot of grammatical and spelling mistakes
Author Response
Response to Reviewer 2 Comments
We sincerely appreciate your careful review of our manuscript and your valuable comments. Those comments are all valuable and helpful for revising and improving our manuscript. We have carefully considered all the comments and made corrections to our manuscript. Here are my revisions and responses to your comments.
Point 1: How the authors determine the deduced genes group?
Response 1: Thank you very much for raising this issue that readers will care about. Our manuscript was not clear about the deduced genes group. The deduced genes group was not determined by us but rather by the FerrDb database team based on current evidence. Our role was to take these genes as our prior information genes for the RF-GSEA method. The relevant content has been added to lines 221 to 229 of the manuscript.
Point 2: All the cohorts are using the same normalization method (log2(TPM+1)?
Response 2: Our manuscript was not clear about the normalization method. Not all cohorts were normalized using the log2(TPM+1) method. Specifically, three RNA-seq cohorts (TCGA-LIHC, LIRI-JP, and IMvigor210) were normalized using log2(TPM+1), while one Affymetrix microarray cohort (GSE116174) was normalized using log2(RMA+1). The relevant content has been added to lines 102 to 104 of the manuscript.
Point 3: How the mixed expression pattern genes are handled? Theoretically, we can't expect all genes will show uniformed expression pattern across patients. How the authors handled those list of genes?
Response 3: The mixed expression pattern genes have always been an important issue in bioinformatics analysis. Our RF-GSEA method can somewhat diminish the impact of mixed expression pattern genes to a certain extent. Firstly, although not all genes exhibit a uniform expression pattern across patients, most genes demonstrate similar expression patterns within patients of the same disease. Genes with mixed expression patterns can be considered as outliers in machine learning. Random forest model employs multiple decision trees, each utilizing only a subset of features from the data, thus, outliers do not significantly impact the model. The utilization of random forest importance scores in subsequent analyses can to some extent handle the mixed expression pattern genes. Secondly, following the acquisition of importance scores, RF-GSEA utilizes GSEA to infer whether gene n is a disulfidptosis-related gene. GSEA facilitates overarching inference based on prior gene information, thereby mitigating the impact of mixed expression pattern genes on the research outcomes. For instance, even if gene n and gene j exhibit distinct expression patterns and are respectively intertwined in close gene regulatory networks with g1, g2, g3, and g4, g5, g6 from the prior gene set, RF-GSEA is still capable of identifying gene n and gene j as disulfidptosis-related genes. Despite our efforts to handle the mixed expression pattern genes, this issue remains unresolved completely. This stands as a primary weakness of RF-GSEA method. The relevant content has been added to lines 628 to 634 of the manuscript.
Point 4: Protein expression level section looks quiet weak and not convincing, it would be better provide more information and validation.
Response 4: The protein expression level section does seem weak. The focus of this study was to develop the RF-GSEA method and validate its effectiveness in HCC. Thus, in this study our main efforts were at the level of data analysis. Although we have validated it as much as possible by analyzing the available data in the main text and supplementary materials. As you said, the protein expression level is not convincing enough. We understand its importance very well, but unfortunately, due to the experimental conditions and resource limitations, we are unable to conduct further experimental validation for the time being. This is also an important flaw of our study. We will endeavor to carry out further validation in future studies. The relevant content has been added to lines 636 to 638 of the manuscript.
Point 5: Figure 3A and B, remove outliers.
Response 5: Figures 3A and B are not clearly described in our manuscript. In fact, the points positioned at the top of the graph are not outliers. These points represent genes with extremely small p-values (<1*10-10) from the GSEA analysis. For the sake of visual clarity, we plotted these genes with a uniform p-value of 1*10-10. The relevant content has been added to lines 338 to 339 of the manuscript.
Point 6: Figure 3E, Set the FC >1.5 and < -1.5 and mark some key genes. Similarly Figure 3F doesn't provide much information, there is no overlap in all four conditions, better provide in some other form.
Response 6: In Figure 3E, the threshold we set was log2(FC)>1 and log2(FC)<-1, which is equivalent to setting FC as >2 and <-2. The setting you recommended, FC >1.5 and <-1.5, is indeed a reasonable choice. However, in this study, our primary aim for conducting the “limma” analysis is to further filter genes obtained from the RF-GSEA method. Different choices of FC may potentially impact our subsequent results. After comprehensive consideration of other literature, we ultimately decided to set FC as >2 and <-2 (Lai G, Zhong X, Liu H, Deng J, Li K, Xie B. A Novel m7G-Related Genes-Based Signature with Prognostic Value and Predictive Ability to Select Patients Responsive to Personalized Treatment Strategies in Bladder Cancer. Cancers (Basel).). As per your request, we've marked some key genes, among which are crucial prior disulfide death-related genes, as well as significant genes for subsequent analysis. Apologies for the lack of a comprehensive explanation regarding the Venn diagram in Figure 3F. Indeed, it encapsulates a wealth of information. Theoretically, in the random forest model, the use of %IncMSE as variable importance scores can lead to negative values. GSEA might successfully enrich due to these negative values, resulting in situations depicted in the following figure. This situation indicates that gene n is highly unlikely to be associated with disulfidptosis. Therefore, we further divided the %IncMSE gene set into two subsets, namely, %IncMSE-Positive (3436 genes) and %IncMSE-Negative (420 genes). %IncMSE-Positive gene set includes genes enriched in the positive direction in GSEA, indicating a high likelihood that these genes are associated with disulfidptosis. %IncMSE-Negative, on the other hand, includes genes enriched in the negative direction in GSEA, indicating that these genes are unlikely to be associated with disulfide death. As anticipated, in the Venn diagram, %IncMSE-Negative gene set has only 20 genes overlap with IncNodePurity gene set and with no gene overlap observed with the DEGs. This demonstrates the robustness of the RF-GSEA method. Due to the segregation of %IncMSE-Negative gene ste from %IncMSE gene set, the four conditions do not overlap. Subsequent analysis only requires the genes overlapping among %IncMSE-Positive gene set, IncNodePurity gene set, and DEGs. The inclusion of Figure 3F not only provides evidence for the robustness of the RF-GSEA method but also elucidates the subsequent selection process of genes undertaken in our study. After thorough consideration, we have opted to persist with the utilization of Figure 3F. We have expounded in greater detail on the contents of this figure in the manuscript, earnestly hoping to gain your approval. The relevant content has been added to lines 289 to 305 of the manuscript.
Point 7: Figure 6, Survival plot, what is the cut point set ? spelling error in OS plot (months).
Response 7: Thank you for your careful review of our manuscript. We have corrected the spelling error in Figure 6 in the manuscript. The cut point was set at a maximum Youden value of 0.381 based on the time-dependent ROC curve in the TCGA-LIHC cohort. The relevant content has been added to lines 411 to 413 of the manuscript.
Point 8: Fig.7 COX model is not clear enough, provide better figure (enlarged form)
Response 8: We enlarged Fig. 7 to provide a clearer presentation of the COX models.
Point 9: Overall strength and weakness of this model should be discussed in discussion.
Response 9: We have added a discussion of model strengths and weaknesses to the manuscript. The relevant content has been added to lines 624 to 638 of the manuscript.
Point 10: Several places, lot of spelling and grammatical errors should be rectified.
Response 10: We've checked our manuscript for spelling and grammatical errors and made corrections.
Reviewer 3 Report
Comments and Suggestions for Authors
In the manuscript „Development of the RF-GSEA Method for Identifying Disulfidptosis-Related Genes and Application in Hepatocellular Carcinoma“ Ni et al. developed a RF-GSEA method in order to identify potential genes related to disulfidptosis. In their study, authors aimed to investigate the relationship between the disulfidptosis-related signature and a HCC prognosis, tumor microenvironment, immunotherapy efficacy, and chemotherapy drug sensitivity. Thus, authors provided evidence of the possible correlation between high-disulfidptosis-related score group and worst patient outcome.
The topic of here presented manuscript is very interesting, and the manuscript is written in a clear way. Although authors didn't provide any in vitro functional analysis of the genes identified in their study to support potential disulfidptosis-related signature, the manuscript still shows intriguing results that will contribute greatly to the field.
Author Response

(The authors gave the same response as above.)

Round 2
Reviewer 2 Report
Comments and Suggestions for Authors
Most of the suggested comments were addressed by the authors.
Comments on the Quality of English LanguageModerate revision required.